EMBO
Molecular Medicine

# Improving the odds of survival: transgenerational effects of infections

Victoria M Spanou [ID][1][✉], Theano P Andriopoulou[1], Evangelos J Giamarellos-Bourboulis [ID][1] &
Mihai G Netea[2,3]

## Abstract

**Recent studies argue for a novel concept of the role of chromatin as a carrier of epigenetic memory through cellular and organismal generations, defining and coordinating gene activity states and physiological functions. Environmental insults, such as exposures to unhealthy diets, smoking, toxic compounds, and infections, can epigenetically reprogram germ-line cells and influence offspring phenotypes. This review focuses on intergenerational and transgenerational epigenetic inheritance in different plants, animal species and humans, presenting the up-to-date evidence and arguments for such effects in light of Darwinian and Lamarckian evolutionary theories. An overview of the epigenetic changes induced by infection or other immune challenges is presented, and how these changes, known as epimutations, contribute to shaping offspring phenotypes. The mechanisms that mediate the transmission of epigenetic alterations via the germline are also discussed. Understanding the relationship between environmental fluctuations, epigenetic changes, resistance, and susceptibility to diseases is critical for unraveling disease etiology and adaptive evolution.**

**Keywords** Epigenetic Memory; Epigenetics; Evolution; Infections;
Transgenerational Epigenetic Inheritance
**Subject Categories** Chromatin, Transcription & Genomics; Evolution &
Ecology; Microbiology, Virology & Host Pathogen Interaction

## Mechanisms of transgenerational transmission of traits: epigenetics

Over the last few decades, recent advances in epigenetic research have dramatically changed our understanding of evolutionary biology. The definition of epigenetics was first described by Conrad Waddington in 1942, as the mechanisms through which the phenotype changes without genotype change (Waddington, 2012). Intergenerational epigenetic inheritance refers to the environmental insults that directly affect the organism and its reproductive cell line, eggs, or sperm by modifying their epigenome, leading to an impact on the phenotype of the first generation of offspring. When these exposures indirectly influence the next generations' phenotypes (second generation and beyond) by transferring the epigenetic imprint, the observed effects are characterized as transgenerational (Fig. 1). Intragenerational and multigenerational are opposite terms referring to the effects occurring within the lifetime of individuals of one generation or involving more than one generation, respectively.

Environmental signals affect the interplay between epigenetics and genomic DNA, resulting in the expression of a specific set of genes regulating cellular function and plasticity (Cavalli and Heard, 2019). Distinct patterns of gene expression define the cellular identity and specialization (Allis and Jenuwein, 2016). The epigenetic mechanisms that dynamically shape the epigenome are essential for cell differentiation and transmission of epigenetic information through cell division. The establishment and maintenance of these expression patterns are tightly interconnected processes which depend on a coordinated set of transcription factors and chromatin effector molecules that bind to specific DNA regions and control the activation or the repression of gene transcription (Allis and Jenuwein, 2016).

This epigenetic regulation takes place at multiple layers, at the transcriptional, post-transcriptional, and post-translational level, with the key modulators including DNA modifications, histone post-translational modifications (PTMs), chromatin remodeling complexes and non-coding RNAs (ncRNAs) (Allis and Jenuwein, 2016; Cavalli and Heard, 2019). For instance, gene expression regulation is mediated by chemical modifications on DNA or histones at the pre-transcriptional level or by micro-RNAs (miRNAs) at the post-transcriptional level. Adding or removing chemical groups, methyl or acetyl groups modifies chromatin structure and architecture by increasing or decreasing the electrostatic binding affinity between DNA and histones, making it more or less condensed. A loosening chromatin folding can be easily accessible to transcription machinery, promoting gene expression while its densely packed form does not enable DNA transcription and results in gene silencing.

In the context of infection, epigenetic regulation orchestrates chromatin architecture and gene expression states in both the host and pathogen. From the host's side, epigenetic processes promote innate immune initiation and activation, leading to pathogen sensing and elimination, while from the pathogen's side, it is

[1]4th Department of Internal Medicine, National and Kapodistrian University of Athens, Medical School, Athens, Greece. [2]Department of Internal Medicine and Radboud Center for Infectious Diseases, Radboud University Nijmegen Medical Centre, 6500HB Nijmegen, the Netherlands. [3]Department of Immunology and Metabolism, Life and Medical Sciences Institute, University of Bonn, Bonn, Germany. ✉E-mail: vimaspanou@med.uoa.gr

**Glossary**

| | | | |
|---|---|---|---|
| **Adaptive evolution** | Evolutionary changes that improve an organism's fitness to respond and adapt to environmental challenges, potentially influenced by epigenetic mechanisms. | | (heterologous) following an initial exposure to a pathogen or pathogen-associated molecules. |
| **Chromatin remodeling** | Dynamic alterations to chromatin structure that affect DNA accessibility, regulate gene expression patterns, and maintain genomic stability. | **Intergenerational epigenetic inheritance** | Epigenetic changes induced by environmental stimuli that directly affect the exposed parental generation (F0, male or non-pregnant female) and their germline cells (F1, sperm or oocyte). In F0, pregnant female individuals, direct exposure affects both F1(unborn offspring) and F2 (offspring germline cells). |
| **CpG islands** | DNA regions with high frequencies of cytosine-phosphate-guanine sequences, usually located near gene promoters and subject to methylation for gene regulation. | **Lamarckian evolution** | A theory of evolution proposing that acquired traits during an organism's lifetime can be inherited by offspring. |
| **Darwinian evolution** | The concept of biological evolution by natural selection supporting the inheritance of the genetic traits that increase the organism's ability to compete, survive, and reproduce. | **Non-coding RNAs (ncRNAs)** | Functionally conserved RNA molecules that do not encode proteins but play a crucial role in regulating gene expression at various levels, including transcriptional, post-translational, and epigenetic regulation. ncRNAs are classified into different categories including micro-RNAs (miRNAs), small interfering RNAs (siRNAs), long non-coding RNAs (lncRNAs), piwi-interacting RNAs (piRNAs), transfer RNAs (tRNAs), ribosomal RNAs (rRNAs), small nuclear RNAs (snRNAs), small nucleolar RNAs (snoRNAs) and enhancer RNAs (eRNAs). |
| **DNA methylation** | A key epigenetic mechanism where methyl groups are added to cytosine residues in CpG dinucleotides, resulting in transcriptional repression or gene silencing. | | |
| **Epialleles** | Heritable variants of a gene caused by epigenetic modifications rather than changes in DNA sequence that may contribute to phenotypic variation. | | |
| **Epigenetic inheritance** | The transmission of epigenetic modifications (e.g., DNA methylation, histone modifications, non-coding RNAs) across generations, shaping offspring phenotype without altering the DNA sequence. | **Quasi-Lamarckian evolution** | A model suggesting that acquired traits, including epigenetic changes, can be inherited through epigenetic mechanisms and play a role in rapid and reversible phenotypic adaptation to fluctuating environments. |
| **Epigenetic memory** | The ability of cells or organisms to retain and transmit gene expression states induced by past experiences, e.g., environmental or developmental signals, without changing the underlying DNA sequence. | **Systemic acquired resistance (SAR)** | A type of immune priming in plants, where exposure to a pathogen induces long-lasting, broad-spectrum resistance to future attacks by the same or a different pathogen. |
| **Epimutations** | Stable, heritable, and potentially reversible changes in gene function caused by modifications to the epigenome, rather than changes in DNA sequence. | **Transgenerational epigenetic inheritance (TEI)** | The transmission of epigenetic marks and phenotypic traits to subsequent generations (F2 and beyond or F3 and beyond in female pregnant individuals) without direct exposure to the environmental stressor. |
| **Germline reprogramming** | A biological process during gametogenesis where most epigenetic marks of germline cells (sperm and oocytes) are erased, ensuring a "reset" of epigenetic information for the next generation. | **Transgenerational immune priming (TGIP)** | The transfer of "immune priming" conferring immune protection and disease resistance across multiple generations, mostly studied in invertebrates. |
| **Histone modifications** | Post-translational modifications of histone proteins, including acetylation, methylation, phosphorylation, and ubiquitylation, that regulate chromatin structure and gene expression. | **Trained immunity** | A form of innate immune memory, characterized by epigenetic and metabolic reprogramming of innate immune cells, induced by a primary exposure to pathogens or stimuli "training" that empower an amplified immune response upon subsequent triggers. |
| **Immune priming** | Enhanced immune response of an organism to a subsequent challenge, either by the same pathogen (homologous) or a different pathogen | | |

involved in pathogen invasion and survival by the suppression of immune responses (Zhang and Cao, 2019). The balanced epigenetic regulation of the immune networks mediating innate immune activation and repression defines the host's susceptibility to infections. Pathogens can modify the pre-existing epigenetic marks on chromatin structure or at specific gene loci leading to disease progression (Baradaran et al, 2019; Zhang and Cao, 2019). Conversely, reversion of these pathogen-induced changes can

promote the host's resistance to infection and contribute to the survival and evolution of the fittest phenotypes (Baradaran et al, 2019).

## DNA methylation

The transfer of methyl groups to cytidine residues in CpGs is catalyzed by epigenetic enzymes called DNA methyltransferases

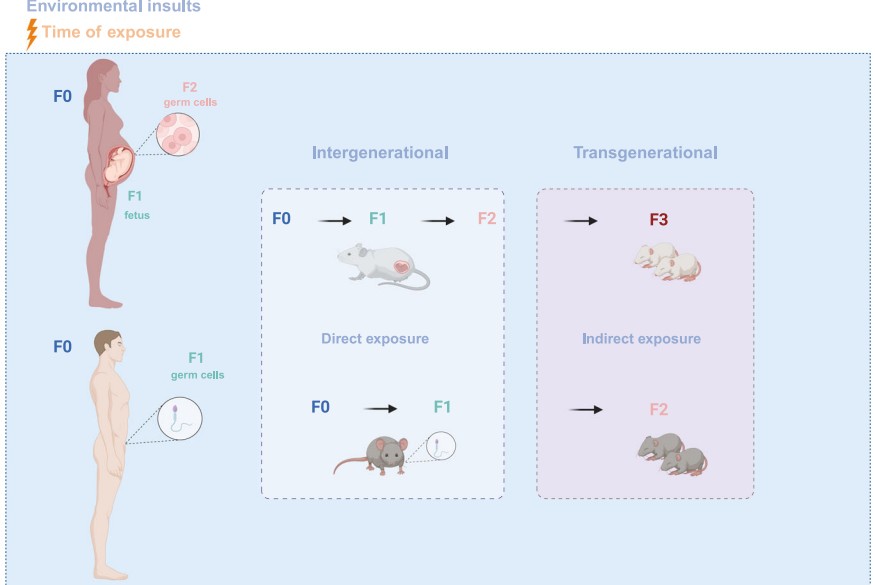

**Figure 1. A schematic overview of intergenerational versus transgenerational epigenetic inheritance via the maternal and the paternal germline.**

Insults of environmental exposures on a mother (F0) during gestation can directly influence the fetus (F1) and its developing germ cells (F2). In the same way, environmental stressors can directly affect the male or female before gestation (F0) and their germline (F1). These phenotypic effects observed after direct exposure represent intergenerational epigenetic inheritance (light blue). Transgenerational epigenetic inheritance represents the phenotypic changes observed on F2 generation offspring or on F3 generation offspring and beyond in the case of a gestating mother induced after indirect exposure to environmental stimuli (light pink).

(DNMTs). The role of DNA methylation across the genome is to sustain the stability and integrity of DNA structure, while methylation in CpG islands of gene promoters renders the gene transcriptionally inactive repressing its expression. These functions are especially important for several biological processes such as cell differentiation, organism development, inactivation of chromosome X in females, and parental imprinting (Moore et al, 2013).

In the context of infection and immune response, in the cotton bollworm *Helicoverpa armigera* larvae, DNMT expression was induced after *Bacillus thuringiensis* infection, while DNMT inhibition led to decreased bacterial growth and increased larvae survival (Baradaran et al, 2019). In the same way, a decreased expression of antimicrobial protein-coding genes was associated with reduced DMNT expression and DNA methylation levels in *G. mellonella* after parasitic infection by *Pimpla turionellae* (Özbek et al, 2020). These studies highlight the important role of DNA methyltransferases in host-pathogen interactions by controlling the expression of antimicrobial peptides.

Interestingly, DNA methylation was associated with the epigenetic inheritance of immune traits, particularly of transgenerational immune priming (TGIP) in invertebrates and trained immunity in mammals, to the next generations (Gegner et al, 2019; Katzmarski et al, 2021). In the tobacco hornworm *Manduca sexta*, TGIP was detected in the larvae of bacterial-challenged parents presenting significant sex-specific alterations in the expression of immune-related genes associated with changes in DNA methylation (Gegner et al, 2019). In mammals, DNA methylation differences were shown in the sperm DNA of male mice that were intravenously infected with *Candida albicans* (Katzmarski et al, 2021). These methylation differences were found in immune gene loci, suggesting a potential epigenetic mechanism underlying the inheritance of trained immunity.

Even more robust evidence for the existence of transgenerational inheritance in mammals was recently revealed, indicating that engineered DNA methylation of promoter-associated CpG islands of metabolism-related genes can be transmitted to the next generation and can specifically become re-established at the epiblast stage (Takahashi et al, 2023). DNA methylation-edited mice presented altered metabolic phenotypes including obesity and hypercholesterolemia induced by gene silencing which were transgenerational transferred, along with the acquired DNA methylation signature, to multiple generations via both maternal and paternal germline (Takahashi et al, 2023). The evidence for epigenetic inheritance of metabolic traits further enhances the possibility of the transmission of infection-induced epigenetic modifications to the offspring regulating their phenotype. However, even though evidence for these processes is substantial, additional independent replications of the findings (including proof of epigenetic editing) are needed.

In plants, cytosine methylation is common across all sequence contexts and often they form epialleles that contribute together to the epimutant phenotype as a quantitative trait loci (QTLs). Small interfering RNAs (siRNAs) trigger a de novo DNA methylation pathway known as RNA-directed DNA methylation (RdDM) (Cooper and Ton, 2022; Fitz-James and Cavalli, 2022; Yang et al, 2022).

## Histone modifications

Chromatin is organized into nucleosomes; each one is formed when 147 base pairs of DNA are tightly wound around eight separate histone protein sub-units, of the core histones H2A, H2B, H3, and H4. Each type of histone protein is subjected to a variety of post-translational modifications including acetylation, methylation,

phosphorylation, and ubiquitylation (Bannister and Kouzarides, 2011). Acetylation and methylation are mediated via histone acetyltransferases (HATs) or deacetylases (HDACs) and methyltransferases (HMTs), respectively, by adding or removing acetyl or methyl groups to lysine and arginine residues on histone tails. Thus, the transfer of acetyl groups on histones by HATs is associated with loosening chromatin folding and increased transcriptional activity while the removal of acetyl groups by HDACs leads to transcriptional silencing.

The vast majority of the epigenetic marks are removed through the extensive epigenetic reprogramming occurring during embryogenesis and spermatogenesis (Hammoud et al, 2009). During spermatogenesis, due to the intense chromatin remodeling, the epigenetic contributions of sperm chromatin to embryo development have traditionally been considered limited compared to maternal contributions. The removal of nucleosomes and replacement with protamines results in the loss of most histone-associated epigenetic marks and the reduction of DNA accessibility for gene expression (Balhorn et al, 1977). However, recent studies challenged this notion: although most of the sperm nucleosomes are evicted, a small fraction of about 1% in mice and 15% in humans are retained (Balhorn et al, 1977; Jung et al, 2017; Tanphaichitr et al, 1978). The presence of residual nucleosomes is a potential mechanism of paternal epigenetic inheritance across generations (Gaspa-Toneu and Peters, 2023).

Epigenetic modifications that escape reprogramming during embryogenesis and spermatogenesis were investigated by employing a transgenic mouse model that overexpresses the histone demethylase KDM1A (Lismer et al, 2020). Using this model, it was shown that modification of histone methylation, specifically histone H3 lysine 4 trimethylation (H3K4me3) via the overexpression of the histone demethylase KDM1A, can escape reprogramming in the embryo and it was associated with transgenerational phenotypes through the paternal germline (Lismer et al, 2020). In the same model, the histone H3 lysine 4 methylation (H3K4me) in sperm was found to be instructive in early embryonic development, regulating gene expression patterns (Siklenka et al, 2015).

In the fruitfly *Drosophila melanogaster*, a stable transgenerational epigenetic transmission by the maintenance of the acquired epigenetic state over multiple generations was demonstrated in generated isogenic *D. melanogaster* epilines that carry alternative epialleles, characterized by persistent differential levels of the Polycomb-dependent histone H3 lysine 27 trimethylation (H3K27me3) mark (Ciabrelli et al, 2017).

In terms of epigenetic inheritance of infection-induced histone modifications, in lipopolysaccharide (LPS)-exposed pregnant rats the transgenerational inheritance of hypertensive phenotype to the offspring was mediated by the inheritable low level of histone H3 lysine 9 demethylation (H3K9me2) (Cao et al, 2022).

## Non-coding RNAs

While only 2% of our DNA encodes for proteins, the rest (98%) can be transcribed to non-coding RNA molecules with their role being under investigation, although mostly is believed to be involved in gene regulation. A large group of non-coding RNA molecules (ncRNAs) have emerged as key regulators of gene expression.

CpG methylation is a well-studied modification of DNA, but it is also observed in various types of RNA, including ncRNAs. It is catalyzed by DNMTs, specifically DNMT2, the most evolutionarily conserved enzyme of the family. DNMT2 acts by modifying RNA molecules, particularly transfer RNAs (tRNAs), and can potentially influence the translation process and protein synthesis. DNMT2 has been involved in paternal TGIP in the red flour beetle *Tribolium castaneu*m (Schulz et al, 2022) and in the paternal transmission of metabolic phenotypes to offspring in mice.

DNMT2-knockdown via RNA interference (RNAi) combined with bacterial priming in adult *T. castaneum* beetles resulted in downregulation and delayed development in the offspring larvae with neither viability nor fertility deficits in the paternal generation. However, paternal knockdown led to increased mortality in the offspring after bacterial infection with *B. thuringiensis*. (Schulz et al, 2022). These results emphasize the important role of DNMT2 in the offspring generation susceptibility, while further research in tRNA methylation in terms of paternal epigenetic inheritance is warranted.

A study by Tyebji et al provides substantial evidence supporting the transgenerational effects of paternal *T. gondii* infection on offspring phenotypes, presenting behavioral changes, and demonstrating that non-coding RNAs play a role in mediating these effects (Tyebji et al, 2020). The study showed the small RNA content of sperm presented significant transcriptional differences between infected and control animals. Microinjection of total small RNA from infected mice into zygotes partially reproduced the behavioral alterations in offspring phenotype (Tyebji et al, 2020). These findings suggest that epigenetic factors in sperm can contribute to the intergenerational transmission of behavioral changes following pathogenic infection, highlighting the potential public health implications of such transgenerational effects. Similarly, more studies in murine models show a causative role of sperm ncRNAs, including long and small ncRNAs, in the intergenerational epigenetic transmission of the phenotypic effects induced by paternal postnatal trauma (Gapp et al, 2020), early trauma (Gapp et al, 2014), chronic stress (Hoffmann et al, 2024; Rodgers et al, 2013) and metabolic disorders (Chen et al, 2016; Zhang et al, 2018).

Within plants, non-coding RNAs such as siRNA and piRNA pathways have also been reported to regulate gene expression (Fitz-James and Cavalli, 2022). Situations of paternal inheritance of siRNAs, which mediate the repetitive establishment of CHH methylation during each replication cycle, have been described. An example of such an inheritance can be observed in *A. thaliana* through male meiocytes, which promotes transposon tameness and gene silence leading to the admission that RNAs are capable of transporting epigenetic messages to the descendants (Fitz-James and Cavalli, 2022).

An overview of the molecular mechanisms reported to be involved in the cross-generational transmission of traits is presented in Fig. 2, while the limited evidence of epigenetic inheritance provided by causation studies is presented in Fig. 3.

# Lamarck and Darwin: two theories of evolution

Inheritance of novel characters acquired during lifetime was the core of the evolutionary theory proposed by Jean-Baptiste Lamarck in his book *Philosophie Zoologique* published in 1809 (Lamarck,

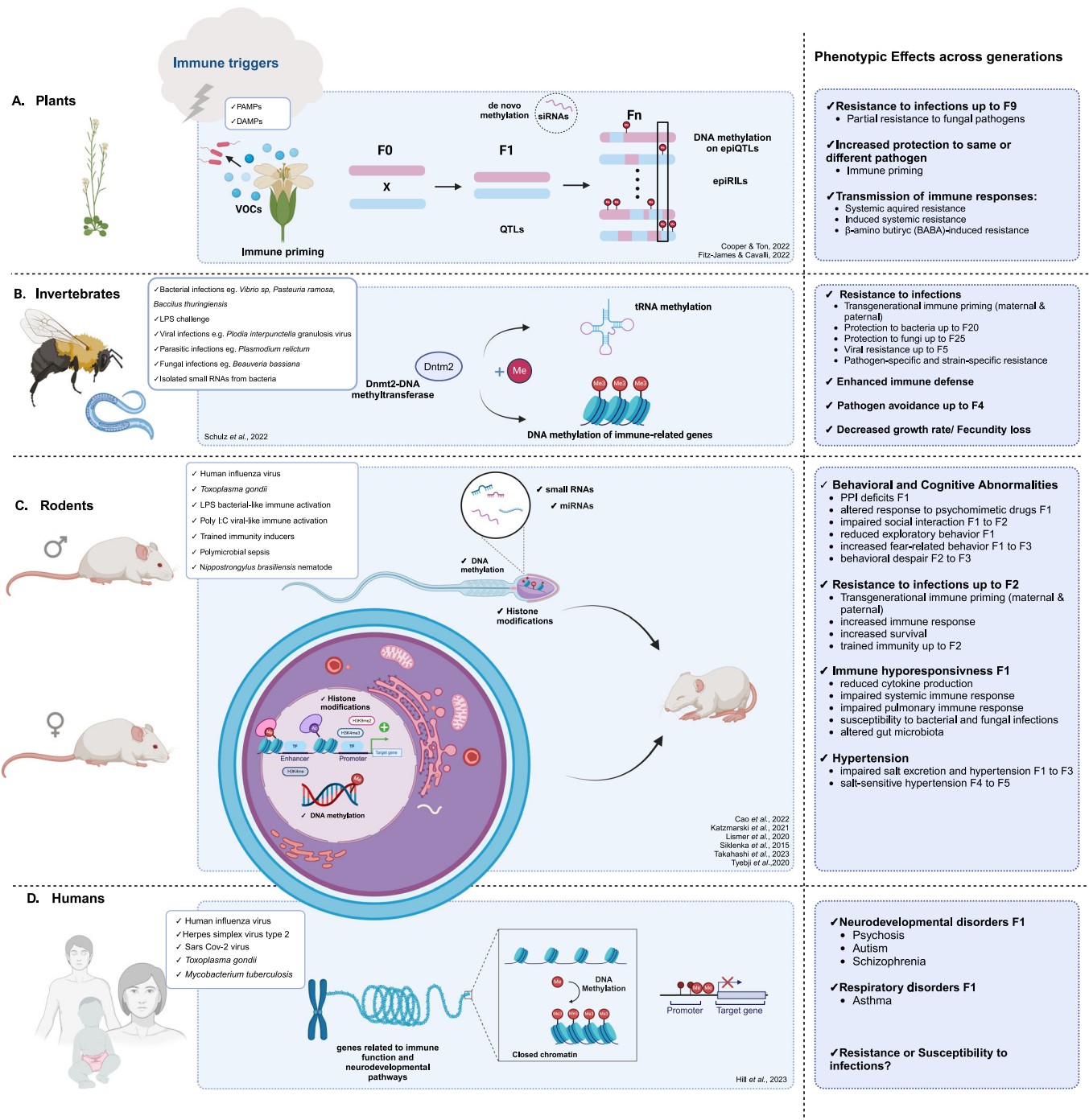

**Figure 2. Epigenetic mechanisms that mediate intergenerational and transgenerational transmission of traits.**

From left to right the Figure shows the different types of organisms, the immune triggers leading to the epigenetic reprogramming, the molecules and mediators involved in the establishment of the epigenetic changes and the transmission of the epigenetic memory to the offspring, and the phenotypic effects observed across multiple generations. (A) In plants, the first line of defense includes Volatile Organic Compounds (VOCs) emission upon Pathogen-Associated Molecular Patterns and/or Damage-Associated Molecular Patterns (DAMPs) engagement. siRNA-guided DNA methylation and chromatin modification contribute to epigenetic regulation and influence gene expression patterns and genome stability. Especially de novo methylation of DNA after exposure to DAMPs or PAMPs is involved in the establishment of epigenetic memory in the next generation. (B) DNA methyltransferase 2 (Dnmt2) is one of the key enzymes of epigenetic regulation in invertebrates. Dnmt2 can lead to immune-related gene methylation and tRNA methylation by the transfer of methyl groups. (C) In rodents, sperm small RNAs are central mediators of the establishment and transfer of epigenetic information. DNA methylation and histone modifications on both egg and sperm are also involved in epigenetic inheritance mechanisms. (D) DNA methylation of genes related to immune function and neurodevelopmental pathways leading to a tightly packed chromatin structure and thus limiting gene expression have been demonstrated in humans after viral infections.

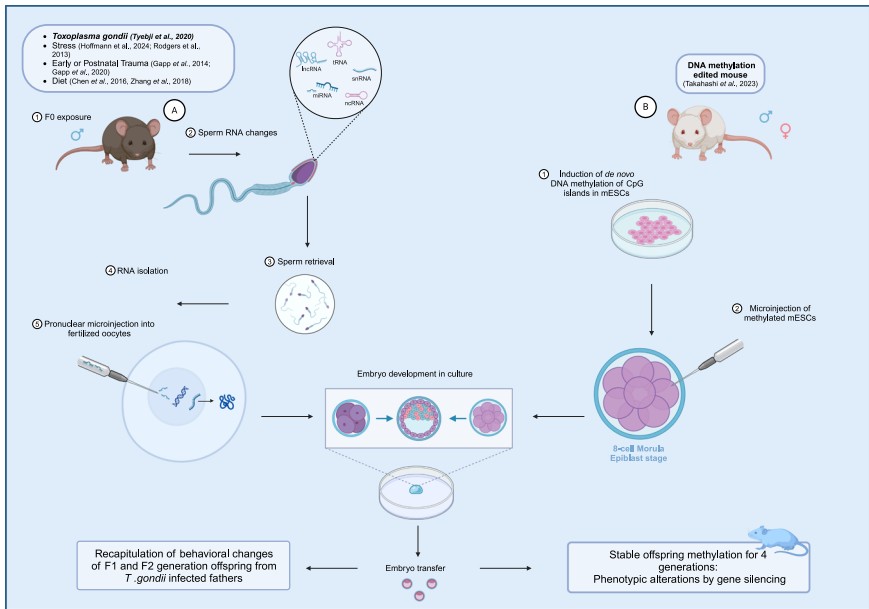

**Figure 3. Epigenetic mechanisms that mediate transgenerational epigenetic inheritance through sperm (small RNAs) and through egg (DNA methylation).**

(A) Male mice experimentally infected with tachyzoites of *Toxoplasma gondii* showed changes in sperm small RNAs. Sperm was retrieved and the total small RNA was isolated and injected into the pronucleus of fertilized eggs using micro-injecting method. Surviving zygotes developed in cell cultures and the embryos were implanted into surrogate mothers. The resulting male and female offspring recapitulated the behavioral changes of first and second-generation offspring from *T. gondii*-infected fathers. (B) A DNA methylation-edited mouse was generated using mouse embryonic stem cells (mESCs), in which targeted CpG islands in promoters of two metabolism-related genes were methylated and silenced. The DNA methylation-edited mESCs were microinjected into 8-cell embryos (epiblast stage) to create a mosaic mouse with normal and methylation-edited cells. The induced DNA methylation and the associated silencing of the specific gene expression stood stably transferred through maternal and paternal germlines across at least 4 generations.

1809). Lamarck argued that organisms can adapt to their environment through their own experiences, and these changes can be inherited by future generations leading to evolutionary progress over time. He proposed that organisms have an innate drive toward complexity and perfection, and they respond to environmental cues by using (or not) certain parts of their body that they continuously develop to acquire new traits.

His view was eclipsed by Charles Darwin's theory of evolution by natural selection, published in 1859 in his book 'On the Origin of Species', which was more evidence-based and rapidly gained widespread acceptance (Darwin, 2011; Tanghe, 2019). Gregor Mendel's work completed Darwin's model by employing it to heredity: how traits pass from parents to their progeny. In the 1950s and 1960s, the milestone discoveries of DNA as vehicle of heritable traits and the subsequent genetic variants as source of inter-individual differences provided a molecular explanation for the Darwinian and Mendelian observations (Arber, 2008; Koonin and Wolf, 2009). DNA was the key molecule, carrier of genetic information across generations, and source of genetic variability for natural selection to act upon. This understanding of how inheritance works through DNA and genes, further discrediting Lamarck's evolutionary theory.

However, a growing body of evidence from the end of the 20th and the beginning of the 21st century, along with the accelerated technological advances in epigenetics support a subtle quasi-Lamarckian model to complement our view on evolutionary processes (Koonin and Wolf, 2009). Technology-based omics tools came to explain the epidemiological and experimental observations of trait inheritance by revealing an upper layer of regulation of the genetic information stored in our DNA (Allis and Jenuwein, 2016). The discovery of epigenetic mechanisms and their role in gene regulation provided strong arguments that information can be stored in DNA beyond the mere sequence of base pairs. Epigenetics has been demonstrated to be the link between the environment and its translation in biological functions.

Epigenetic remodeling as a reaction to environmental signals allows phenotypic modification without altering the genotype and accounts for a large proportion of the variability of host traits within a population (Fox et al, 2019). Epigenetic marks established after environmental fluctuations have the potential to perpetuate gene expression patterns over long periods of time ensuring the phenotypic adaptation under stress conditions. Increasing evidence argues that the persistence of the epigenetic landscape of specific traits, named epialleles, can serve as a background in which selection can act. In plants, for example, phenotypes are dynamically adjusted by epigenetic changes that occasionally are established in specific genomic loci and inherited steadily as epialleles, with a potentially beneficial role in their adaptation and evolution (Srikant and Tri Wibowo, 2021). However, the mechanisms of heredity of the non-genetic information and their role in adaptive evolution remain largely unexplored in mammals in general, and humans in particular.

## Evolutionary arguments for transgenerational effects

While the Darwinian evolutionary model takes place over long periods, with only the most adaptable to the environment phenotype surviving forced by natural selection, epigenetic inheritance following a quasi-Lamarckian model is more advantageous for a quick and reversible adaptation (Koonin and Wolf, 2009). Epigenetic inheritance contributes to an organism's temporary survival by employing a "bet-hedging" strategy of returning to the original epigenetic state following environmental fluctuations.

However, a challenging question is whether quick adaptation and reversibility of the phenotype (plasticity) is a new evolutionary strategy or whether it slows up species evolution by natural selection (Fox et al, 2019). An additional important point to make is that epigenetic inheritance that mediates phenotypical changes by conferring the organism and its offspring a survival advantage can also result in some maladaptive effects, as reported in studies from lower organisms to humans.

Epidemiological and immunological data show that exposures to environmental factors such as unhealthy diets or famines, stress, toxins, smoking, cold temperature, and infections can affect the organism at any time during its lifetime leading to changes in its epigenome (Pembrey et al, 2014; Rowe and Rowe, 2008; Sun et al, 2018; Svanes et al, 2017). These alterations are especially important when established in sensitive time windows of growth, like pregnancy, early life, and puberty, as they can become imprinted on the germline cells (Cecilie et al, 2022; Pembrey et al, 2014; Svanes et al, 2017). When germ cells undergo development, they experience a process called reprogramming, during which most epigenetic marks (represented mainly by DNA methylation) are erased, a process shown mainly in mammals. However, certain epigenetic marks, including DNA methylation patterns, can escape reprogramming. Through the germline, epialleles containing the sustained epigenetic information can pass down from one generation to the next and influence the offspring phenotype.

Importantly, negative long-term consequences on the next generations have been also epidemiologically and experimentally observed. Human epidemiological studies show that parental environment, particularly the father's exposures to smoking or overweight years before conception, may play a key role in asthma and lung function in the children (Cecilie et al, 2022; Pembrey et al, 2014). Of note, a critical window of susceptibility during male prepuberty has been identified (Cecilie et al, 2022). Mounting evidence in animal models also shows that inherited epigenetic modifications induced after parental exposure to high-fat diets can reprogram the next generation's phenotypes, promoting the development of neurodevelopmental and metabolic disorders, including type 2 diabetes (Bodden et al, 2020). Recent discoveries in rodents reveal that altered sperm epigenome after environmental insults can deliver the epigenetic information to the oocyte at fertilization and hence modulate offspring susceptibility to diseases. Diet-induced alterations in sperm can influence the metabolism and brain function of the progeny. In particular, sperm non-coding RNAs and DNA methylome has been claimed to mediate these effects through the paternal germline. However, a causative role of these in vivo epigenetic editing processes remains to be demonstrated.

Intergenerational and transgenerational effects have been reported in the progeny of humans who experienced major environmental distresses during the Second World War (Yehuda et al, 2005), including famines (the Dutch famine occurred in 1944–1945 and the Chinese Great Famine occurred in 1959–1961) (Shen et al, 2019) and Holocaust, with their children presenting increased stress susceptibility and metabolic disorders like dyslipidemia. Epigenetic changes like alterations in methylation levels in specific gene loci have been associated with these exposures and their impact on offspring phenotypes implying the existence of mechanisms of epigenetic inheritance (Shen et al, 2019). One of the most consequential challenges the world faced in the last years was the COVID-19 pandemic, leaving a huge burden of both physical and mental health issues with long-term and potential transgenerational consequences through epigenetic inheritance (Kleeman et al, 2022). It is of vital importance to investigate the impact of infections on the next generations and shed light on the epigenetic mechanisms whereby infections may lead to a more resistant or susceptible phenotype. An overview of the organisms in which intergenerational and/or transgenerational effects of infections have been documented is presented in Fig. 4 and a summary of the literature is presented in Table EV1.

## Experimental arguments for transgenerational effects of infections in invertebrates

In nature, hosts encounter a wide range of pathogens including parasites, bacteria, fungi, and viruses, and shape their immune defense to ensure protection and resistance to infections. Host-pathogen interactions can impact host life history, and as a result, have marked effects on community structure and evolutionary dynamics over time. Populations occupying a certain habitat are more likely to be repeatedly exposed to the same pathogens, either within the same or across the next generations (Moret and Siva-Jothy, 2003).

Invertebrates were long assumed to lack immunological memory developed after encounters with pathogens, which is naturally seen in vertebrates (Little and Kraaijeveld, 2004). This was assumed due to the absence of an adaptive immune system in invertebrates, which rely solely on innate immune host defense mechanisms. However, increasing evidence over the years revealed that invertebrates' immune defense is far more complex than previously considered. Invertebrates develop defense mechanisms against pathogens under persistent selective pressure, which can be transgenerationally transferred over generations to afford protection (Little and Kraaijeveld, 2004). Following initial exposure to a pathogen, an increased protection termed "immune priming" is elicited after a secondary challenge with the same or a different pathogen (Tetreau et al, 2019). Moret and Silva-Jothy showed that a survival advantage to reinfections was provided to the beetle *Tenebrio molitor* after a previous infection experience, as an adaptation strategy (Moret, 2006; Moret and Siva-Jothy, 2003). A cross-protective immune response against fungal pathogens was evidenced in the mealworm beetle *Tenebrio molitor* which had been previously exposed to LPS (Moret, 2006; Moret and Siva-Jothy, 2003). However, even though there are cases in which this protection is broad, the more pronounced effect is exhibited on

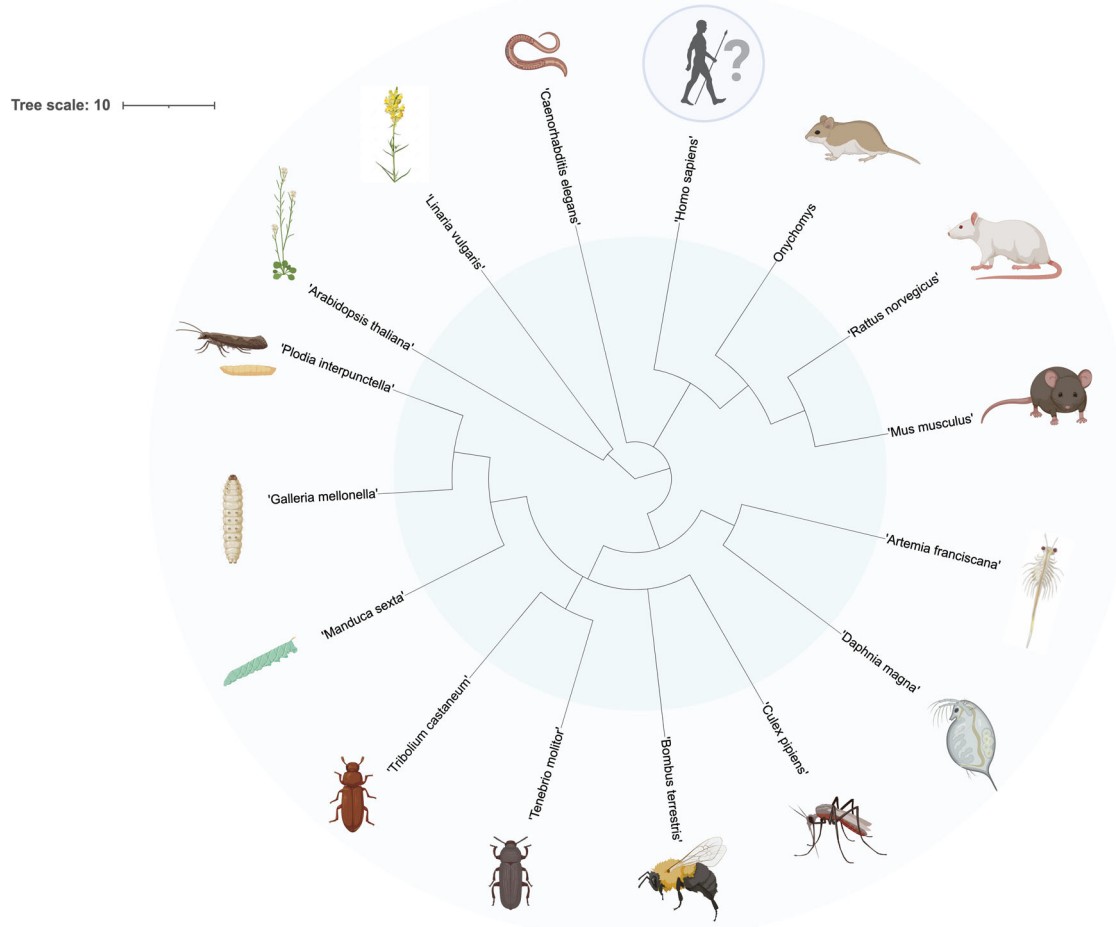

**Figure 4. Overview of the species phylogeny of the organisms in which intergenerational and/or transgenerational effects have been demonstrated.**

The phylogenetic tree includes 2 plant species: *Linaria vulgaris* and *Arabidopsis thaliana*, 10 invertebrate species: *Plodia interpunctella, Galleria mellonella, Manduca sexta, Tribolium castaneum, Tenebrio molitor, Bombus terrestris, Culex pipiens, Daphnia magna, Artemia franciscana* and *Caenorhabditis elegans,* and 4 vertebrate species: *Mus musculus, Rattus norvegicus, Onychomys* and *Homo sapiens.*

the pathogen that induced the immune priming (Roth et al, 2009; Sadd and Schmid-Hempel, 2007).

A growing body of data in invertebrates supports that this innate immune memory and protection can be transgenerationally transmitted to the progeny, a biological process referred to as "TGIP" (Little and Kraaijeveld, 2004). The afforded protection to the offspring seems to be pathogen-specific, as the progeny from parents exposed to a specific pathogen tend to be more resistant to infections induced by the same pathogen than to heterologous pathogens (Roth et al, 2009). A more pronounced protection was evidenced in the brine shrimp *Artemia franciscana* subjected to the homologous *Vibrio* strain infection for which their parental generation was primed compared to a heterologous *Vibrio* strain (Roy et al, 2022). Similarly, strain-specific immunity was reported in the crustacean *Daphnia magna* exhibited more enhanced immune protection when exposed to the bacterial strain *Pasteuria ramosa* that their mother had encountered than to a different strain (Little et al, 2003). In honey bees *Apis mellifera*, induction of social immunity through TGIP against *Paenibacillus larvae* infection via the vitellogenin-hypopharyngeal gland axis was demonstrated upon feeding the nurse bees with heat-killed pathogens *Ascosphaera apis* spores (fungi) and *Paenibacillus larvae* cells (bacteria) (Kim et al, 2023). Immune memory against viral infections was also proven in *Plodia interpunctella* exposed to its natural DNA virus after a secondary viral challenge, affording both intra-generational and transgenerational protection (Tidbury et al, 2011). In *Aedes aegypti* mosquitos, transgenerational antiviral immunity transferred up to F3 generation offspring was observed after maternal infection with three different RNA arboviruses through the vertical transfer of integrated viral DNA sequences (Rodriguez-Andres et al, 2024). Mondotte et al further showed that this antiviral transgenerational priming with RNA viruses, in *Aedes aegypti* mosquitos and *Drosophila melanogaster* flies is sequence specific and can last up to 5 generations of offspring (Mondotte et al, 2020).

While most studies focus on the effects of maternal TGIP, paternal life experiences' "fingerprints" can also be transmitted to the next generations and influence their phenotype. Increasing evidence supports the existence of both female and male-induced TGIP (Roth et al, 2010; Roth et al, 2009). Roth et al showed that in the red flour beetle, *Tribolium castaneum*, paternal immune priming with heat-killed bacteria can transfer resistance to their

offspring as well (Roth et al, 2010; Roth et al, 2009). In the same way, exposure of adult male beetles *T. castaneum* to a non-lethal dose of bacteria protects their offspring against a potentially lethal dose of the same pathogen (Schulz et al, 2022). The afforded protection is more pronounced when offspring are exposed to the same bacteria as their parents (Roth et al, 2010; Schulz et al, 2022).

While paternal TGIP has been observed, the mechanism behind this transfer of protection is not yet fully understood. Eggert et al showed in a double mating experiment that only the genetic offspring of immune-primed males exhibited enhanced survival when exposed to bacterial pathogens, suggesting that TGIP is mediated by sperm and potentially via its epigenome (Eggert et al, 2014). However, phenoloxidase activity, a key immune trait in insects, was increased in both biological and step offspring, suggesting that mediators transferred via seminal fluids may have additional effects on offspring's immune traits (Eggert et al, 2014). These findings provide insights into the mechanisms underlying paternal transgenerational immune priming and highlight the potential roles of sperm and seminal fluids in shaping offspring immune responses.

Another transgenerationally inherited behavioral trait employed by *Caenorhabditis elegans* is pathogen avoidance, which can distinguish pathogenic from beneficial bacterial food and increase survival in its environment. Exposure to pathogenic *Pseudomonas aeruginosa* induced pathogen avoidance in four subsequent generations through either male or female germline (Moore et al, 2019). A decreased survival variability among generations without an overall change in survival was revealed after exposure of a primed lineage of *C. elegans* to *Salmonella enterica* associated with modifications in gene expression (Wibisono and Sun, 2023). Interestingly, the control lineage displayed a repeating pattern of survival that persisted over 12 generations (Wibisono and Sun, 2023).

Several mechanistic studies explore the epigenetic mechanisms that mediate the transgenerational inheritance of this learned behavior in *C. elegans*. Moore et al showed the involvement of Piwi/Argonaute pathway and TGF-β in sensory neurons (Moore et al, 2019). Additional studies showed that purified small RNAs isolated from different *Pseudomonas sp.* including the ncRNA P11 from *P. aeruginosa* (Kaletsky et al, 2020), the small RNA Pv1 expressed by *P. vravovensis* (Sengupta et al, 2024) and the small RNA Pfs1 isolated from *P. fluorescens* (Seto et al, 2024), are both necessary and sufficient to induce pathogen avoidance up to F4 generation offspring. Further findings reveal the essential role of *Cer1* retrotransposon in "reading" bacterial small RNAs required for the learning and inheritance of RNA-induced pathogen avoidance (Moore et al, 2021).

While transgenerational effects of infections may improve the fitness of the progeny via TGIP, substantial data refer to the collateral transfer of important costs. In the red flour beetle *Tenebrio molitor* challenged with heat-killed bacteria, maternal TGIP decreased the developmental time of the offspring, whereas paternal TGIP reduced offspring fecundity (Roth et al, 2010). Pigeault et al evaluated the costs and benefits of the transgenerational impact of a mother's infection in *Culex pipiens* mosquitos and showed that the infected offspring of *Plasmodium*-infected mothers presented considerable fecundity loss, 3 times higher than the infected offspring of non-infected mothers in which no protection to infection was transmitted (Pigeault et al, 2015). Similarly, a drop in reproduction capacity associated with the reduction of insemination rate in infected females and a decline of progeny fitness, size, and survival was observed in *Anopheles coluzzi* malaria vector mosquitos upon parental infection with the entomopathogenic bacteria *Chromobacterium anopheles* (Gnambani et al, 2023). In the same way, maternal immune priming of *Aedes aegypti* mosquitos with different serotypes of Dengue virus resulted in a trade-off in the life history of their daughters, changes of pupation time, and shift in sex ratio towards females being more prevalent when their mothers experienced a heterologous infection (Cime-Castillo et al, 2023). In this case, there was still a protective effect with the offspring of primed mothers being more resistant to Dengue viral infection (Cime-Castillo et al, 2023). These results are of particular interest, as mosquito vectors have the potential of parasite harboring and transmission of diseases to humans and animals. On the contrary, resistant generations of the wax moth *Galleria mellonela* to *Bacillus thuringiensis* displayed a positive trade-off, with larger pupal mass and increased fecundity compared to non-resistant lines (Dubovskiy et al, 2016). This feature is exceptional since most of the studies refer to the negative trade-offs caused by the evolutionary forces, for example, decreased growth rate and fecundity.

Invertebrates employ humoral and cellular responses to fight pathogens. Humoral immunity involves the releasing of immune factors that are either toxic for the pathogens (antimicrobial peptides) or enhance cell-mediated pathogen clearance (phenol oxidase, complement). Cellular immunity is elicited following pathogen recognition via the pattern recognition receptors (PRRs) by cell-mediated responses such as phagocytosis and encapsulation/melanization (Melillo et al, 2018). For instance, male offspring derived from LPS-challenged *Bombus terrestris* bumblebees showed enhanced immune defense associated with higher phenol oxidase activity which promotes pathogen clearance via encapsulation and melanization (Moret and Schmid-Hempel, 2001). In addition, LPS challenge of the mealworm beetle *Tenebrio molitor* promoted their offspring resistance by the secretion of antimicrobials peptide in the hemolymph (Moret, 2006).

The larvae of the wax moth *Galleria mellonella*, showed increased resistance to the pathogenic fungus *Beauveria bassiana* by modifying its immune response up to the 25th generation, suppressing the systemic immunity while enhancing a more targeted cuticular immune defense since the cuticle is the most important barrier for fungal invasion (Dubovskiy et al, 2013). This enhancement of insect's immune defense was mediated by the upregulation of antimicrobial peptides (e.g., galiomicin) and stress/immunity-related genes (e.g., insect metalloproteinase inhibitor-IMPI) cuticular expression, as long as by the increased phenol oxidase activity in the integument. More recent data demonstrate the involvement of epigenetic mechanisms in the regulation of both the invertebrates' immune defense and the TGIP through the inheritance of the epigenetic marks from the parents to their offspring (Gegner et al, 2019; Mukherjee and Dobrindt, 2022).

# Transgenerational effects of infections in plants

Not only invertebrate animals show transgenerational transmission of resistance to infections. Plants are complex organisms with ecological and economic significance. Numerous experimental data

have shown that DNA demethylation of epialleles is the key to epimutation and developmental epigenetics in plants (Fitz-James and Cavalli, 2022). They exhibit a wide variation within the same species and genome, with an absence of reset through generations, amplifying the impact of transgenerational epigenetics in plants (Quadrana and Colot, 2016). In addition to the developmental epigenetics and the transgenerational inheritance of phenotypes, such as the variation of Lcyc locus in the *Linaria vulgaris* species due to naturally occurring DNA methylation, an increasing number of studies reported the transgenerational inheritance of defense and immune priming in plants (Quadrana and Colot, 2016).

During an infection, microbial-associated molecular patterns (MAMPS) and/or damage-associated molecular patterns (DAMPs) activate the innate receptors leading to the activation of innate immunity. Another immune response, effector-triggered immunity (ETI), is evoked by the pathogen effectors that play an immunosuppressive role (Cooper and Ton, 2022; Espinas et al, 2016). Immune priming offers increased protection after a secondary challenge with the same or a different alarm signal, but many studies show that this phenomenon is mostly reversible rather than transgenerationally transmissible. However, cases have been described in which immune priming can be passed down through generations, with certain studies suggesting the possibility of priming transmission through volatile organic compounds (VOCs) (Cooper and Ton, 2022). Three major priming responses have been mostly studied, in an attempt to gain more understanding, overcome the hurdle of variated data, and elucidate some of the fundamental mechanisms involved: systemic acquired resistance (SAR), induced systemic resistance (ISR), and β-amino butyric (BABA)-induced resistance (BABA-IR) (Yang et al, 2022).

Transgenerational inheritance of these immune responses in plants has been mentioned in many studies, such as in the case of *A. thaliana* where cytosine methylation provided pathogen resistance for nine generations (Fitz-James and Cavalli, 2022). In addition, DNA methylation in several epiQTLs (quantitative trait loci) gave *A. thaliana* partial resistance to fungal pathogens (Fitz-James and Cavalli, 2022; Quadrana and Colot, 2016; Yadav et al, 2018), indicating immune priming.

However, not all studies demonstrated cross-generational transmission of resistance. An experimental study of the standard ecotype of *A. thaliana* failed to prove that defense priming in parental plants after exposure to *Pseudomonas syringae* pv. tomato passes as an epigenetic characteristic to the descendants (Yun et al, 2022). Given the wide range of data and outcomes in many studies in the same species, additional studies on the transgenerational inheritance in plants are needed.

# Transgenerational effects of infections in mammals

## Intergenerational effects

An increasing amount of evidence from epidemiological and genome-wide association (GWAS) studies in humans and experimental data in animal models argues that epigenetic marks induced by immune triggers and underlying DNA sequences can be passed down to subsequent generations and influence their phenotypes.

Human maternal exposure to infection or inflammation during pregnancy has been suggested as a risk factor for neurodevelopmental and psychiatric illnesses like schizophrenia, autism, and bipolar disorder in the offspring [46]. Epidemiological studies show that higher winter infectious diseases incidence such as pneumonia and influenza was associated with higher rates of schizophrenia in the progeny (Watson et al, 1984). Likewise, data from human case-control studies show that maternal exposure to infections during pregnancy including influenza (Brown et al, 2004), herpes simplex virus type 2 (Buka et al, 2001), and *Toxoplasma gondii* (Brown et al, 2005) during pregnancy increased the risk for the development of schizophrenia and psychosis in offspring. Follow-up data from the Respiratory Health in Northern Europe (RHINE) study assessing the effects of maternal and/or paternal tuberculosis in their offspring found a high risk of asthma, with an adjusted odds ratio (OR) of 1.29 [95% confidence interval (CI), 1.05–1.57] in offspring of parents with tuberculosis and a significant association with an adjusted OR of 1.58 [95% CI, 1.29–2.05] of parental tuberculosis with allergic asthma (Gyawali et al, 2023). Likewise, the risk for asthma was higher in individuals with parental tuberculosis in childhood OR 1.73 [95% CI, 1.20–2.50] or later preconception OR 1.38 [95% CI, 1.00–1.91], than in the ones with parental TB after offspring's birth. These differences were significant only in the maternal line in childhood OR 1.95 [95% CI, 1.13–3.37] and in later preconception OR 1.74 [95% CI, 1.08–2.80] (López-Cervantes et al, 2022) periods.

The potential for long-term and multigenerational health effects arising after a pandemic was evidenced in epidemiological studies from the Spanish influenza pandemic in 1918–1919 (Almond, 2006; Almond and Mazumder, 2005; Fletcher, 2018). Children who experienced in utero exposure to pandemic influenza exhibited decreased educational attainment, lower socioeconomic status (Almond, 2006; Fletcher, 2018), and increased incidence of a range of health problems including diabetes, stroke, hypertension, kidney dysfunction, and cancer associated with the time of the fetal exposure (Almond and Mazumder, 2005). Recently, the COVID-19 pandemic affected globally more than 700 million individuals of different age groups (WHO COVID-19 Dashboard. Geneva: World Health Organization, 2020. Available online: https://covid19.who.int/ (last cited: [21.06.2023]), leaving a huge burden of short-term and long-term economic, mental, and physical outcomes. In a retrospective cohort study, maternal SARS-CoV-2 infection was associated with neurodevelopmental disorders in offspring in the first year of their life with an adjusted OR of 1.86 [95% CI, 1.03–3.36]; $p = 0.04$ (Edlow et al, 2022), although it is not clear yet whether this is mediated via biological or social mechanisms.

The novel diagnostic tools allowed for deepening our knowledge of the mechanisms that govern the intergenerational effects in humans. Some studies in humans have indicated the potential for epigenetic inheritance in individuals exposed to prenatal COVID-19 infection exhibiting altered DNA methylation patterns in genes related to immune functions in both the exposed individuals and their offspring (Corley et al, 2021; Hill et al, 2023; Ozturkler and Kalkan, 2022). A distinct pattern of hypermethylated IFN-γ genes and hypomethylated inflammatory genes was revealed in the peripheral blood mononuclear cells of critically ill COVID-19 patients (Corley et al, 2021). In addition, genome-wide analysis in infants' specimens in utero exposed to SARS-CoV-2 showed altered

DNA methylation patterns of genes, associated with human neurodevelopmental pathways (Hill et al, 2023). However, mechanistic studies in humans are limited and more research enrolling larger sample sizes for longer follow-up across the next generations is needed to validate these findings and exclude confounding factors. A critical point that needs to be emphasized and ruled out is the possible impact of direct effects of infection and "vertical transmission" in the fetus. Therefore, the research focusing on the epigenetic inheritance through the paternal germline to the offspring is often a stronger argument for the actual epigenetic transmission.

The understanding of the biological and pathophysiological processes underlying these effects comes from animal studies indicating the possible mechanisms whereby infections lead to the phenotypic changes of the offspring. Different studies in translational rodent models have shown that parental infections, such as the influenza virus (Shi et al, 2003), *Toxoplasma gondii* (Tyebji et al, 2020), and polymicrobial sepsis (Bomans et al, 2018), or inflammation (Cao et al, 2022; Tschurtschenthaler et al, 2016; Weber-Stadlbauer et al, 2017) can epigenetically reprogram germline cells and thus affect phenotypes and disease outcomes in offspring. In particular, prenatal exposure of mice to the human influenza virus leads to long-term deleterious effects in the brain development of their progeny, mainly atrophy of pyramidal cells and macrocephaly associated with behavioral anomalies that persist until the adult age (Fatemi et al, 2002). Maternal infection during pregnancy, with the human influenza virus, can lead to abnormal behavioral responses in offspring, that resemble some characteristics seen in schizophrenia and autism such as deficits in prepulse inhibition (PPI) (Shi et al, 2003). Also, mice born to influenza-infected mothers displayed altered responses to antipsychotic and psychomimetic drugs, reduced exploratory behavior, and impaired social interaction (Shi et al, 2003).

Intriguingly, the maternal immune response itself can have a profound effect on these behavioral changes. Studies using the synthetic double-stranded RNA polyinosinic-polycytidylic acid (poly I:C) which mimics the viral-induced immune response, or the LPS which mimics bacterial-induced systemic inflammation, showed that maternal immune activation alone can result in PPI deficits or hypertension in the offspring, respectively (Cao et al, 2022; Shi et al, 2003). Likewise, Weber-Stadlbauer et al showed that the F1 and F2 generation offspring of immune-challenged with poly I:C mothers during pregnancy exhibited impaired social interaction and increased fear-related behavior, indicating the long-term outcomes of prenatal immune insults that persist intergenerationally (Weber-Stadlbauer, 2017). These experimental data show that prenatal immune activation alone can lead to the development of immune-triggered brain abnormalities across generations, suggesting potential links to neurodevelopmental psychiatric disorders, such as schizophrenia, autism, and bipolar disorder mediated via non-genetic inheritance mechanisms.

Despite the extensive research on the effects of maternal immune activation or infection during pregnancy on offspring health, the impact of paternal contributions is largely unexplored. Interestingly, the above-mentioned behavioral alterations induced after in utero poly I:C exposure were presented in case of males only in paternal lineage-derived F2 generation offspring, which also exhibited a new phenotype of behavioral despair (Weber-Stadlbauer, 2017). Behavioral changes were also observed in the

offspring of *Toxoplasma*-infected fathers and these changes were displayed in a sex-dependent manner (Tyebji et al, 2020). Epigenetic mechanisms including the sperm small RNAs are involved in the behavioral inheritance since the behavioral phenotypes were partially reproduced after the injection of sperm small RNAs of infected mice into zygotes, providing substantial evidence of the contribution of sperm epigenetic factors to the intergenerational transmission (Tyebji et al, 2020).

Paternal non-genetic inheritance was also investigated in the offspring of male mice subjected to sepsis using the cecum ligation and puncture model (Bomans et al, 2018). The descendants of septic fathers presented with a reduced body weight in a sex-specific manner, with the male offspring displaying more pronounced effects than females. Male offspring of septic fathers showed a lower cytokine response in plasma and lower TNF-α production by their alveolar macrophages following stimulation of zymosan, a fungal component. This immune hypo-responsiveness presented with an impaired systemic and pulmonary immune response in the male progeny from septic fathers may increase their risk for bacterial and fungal infections (Bomans et al, 2018). Sperm methylome analysis revealed that paternal sepsis-induced changes in genes regulating the developmental, metabolic, and biosynthetic pathways may penetrate and modify the offspring's phenotype. In the same way, the transmission of inflammatory traits via alterations in DNA methylation signature of metabolism-related genes, like *Igf1r* and *Nr4a2*, was evidenced in sperm and epithelial cells of F1 offspring of male mice subjected to experimental colitis, associated with decreased body weight, and increased susceptibility to colitis (Tschurtschenthaler et al, 2016).

## Transgenerational effects

The transmission of the intergenerational effects of maternal immune exposures on first and second-generation offspring or of paternal exposures on their direct offspring is supported mostly in rodents but also in humans. However, evidence of transgenerational inheritance of phenotypic changes across subsequent generations without being directly affected at the time of the infection or immune activation is very limited in animal models and still lacking in humans.

Persistence of altered behavioral phenotypes in F3 generation offspring was demonstrated in a mouse model of maternal prenatal immune activation with poly I:C (Weber-Stadlbauer, 2017). F3 murine offspring through the paternal linage presented behavioral changes such as behavioral despair, impaired social interaction, and increased cued fear expression indicating that transgenerational inheritance is mediated via the paternal lineage (Weber-Stadlbauer, 2017). Behavioral despair was a novel phenotype, that emerged only in the F2 and F3 offspring, skipping the direct offspring, which shows that maternal immune activation may induce latent behavioral manifestations in subsequent generations. Analogously, paternal pre-conceptual viral-like immune challenge by poly I:C led to brain and behavioral alterations in their offspring which were transferred up to the F2 generational offspring (Kleeman et al, 2024). Sperm small non-coding RNA analysis revealed significant changes in paternal sperm microRNA content at the conception time and of the sperm PIWI-interacting RNA content of their male offspring (Kleeman et al, 2024). Similarly, *T. gondii*-infected fathers transmitted the altered behavioral phenotypes until the F2

generation by epigenetic inheritance of the small RNA changes induced in their sperm during infection (Tyebji et al, 2020). However, in contrast to the previous studies, only the latter provided a causative link and robust evidence that non-coding RNAs contribution to paternal epigenetic inheritance is substantial.

Transgenerational inheritance of hypertension across five generations was also evidenced in the offspring of LPS-challenged pregnant rats (Cao et al, 2022). Specifically, impaired salt excretion and hypertension were observed in the first to third generations, with the fourth and fifth generations exhibiting salt-sensitive hypertension. Once LPS-challenged pregnant rats were treated with tempol, a reactive oxygen species scavenger, hypertension was successfully prevented in the first-generation offspring as well as its transgenerational inheritance (Cao et al, 2022).

Overall, the transgenerational effects of infections in humans are still under debate. Given the huge burden of health and socio-economic consequences that arose after the COVID-19 pandemic affecting almost the total global population and the high risk of emerging infectious diseases, infections are at the center of the main interest. Intergenerational effects have already been described in COVID-19, associated with the development of mental disorders in the offspring. Altered DNA methylation patterns of several genes implicated in immune and neurodevelopmental processes have been revealed in the direct offspring (Corley et al, 2021; Hill et al, 2023; Ozturkler and Kalkan, 2022). The potential transmission of these epigenetic changes to the subsequent generations and how they may influence their phenotype is crucial to be investigated to provide substantial evidence for the existence of transgenerational inheritance of infections in humans.

### Trained immunity: an innate immunological memory of infections

Immunological memory built-up after an infectious insult confers the organism with a new weapon to fight against disease by adapting its immune system. Until recently, immunological memory was thought to be an exclusive characteristic of adaptive immunity, however, a growing body of data shows that innate immunity can also display a de facto innate immune memory termed *trained immunity* (Netea et al, 2020). After the first stimulus, innate immune cells and their progenitors in the bone marrow undergo epigenetic and metabolic remodeling, empowering them to develop an augmented immune response after a secondary heterologous stimulus. This epigenetic reprogramming is known to enhance the expression of certain immune and metabolic genes providing a better antimicrobial ability to the host cells to resist infections.

While the phenomenon of trained immunity has been previously demonstrated in plants and invertebrates, compelling evidence for the transgenerational transmission of trained immunity across generations was recently provided in mammals (Katzmarski et al, 2021). Offspring of fathers subjected to a sublethal systemic infection with *Candida albicans* or a challenge with zymosan for the induction of trained immunity exhibited improved immune responsiveness and enhanced protection against systemic infections caused by heterologous pathogens, such as *Escherichia coli* and *Listeria monocytogenes*. These findings have implications for our understanding of immune system regulation highlighting the potential transfer of innate immune memory to subsequent generations (Katzmarski et al, 2021). Additionally, a newly proposed concept suggests the induction of trained immunity during pregnancy, by the priming and training of decidual natural killer cells which contribute to their improved

functionality and protection against infections (Feyaerts et al, 2021). In a healthy pregnancy, a controlled level of inflammation is necessary for successful implantation, placental development, and immune tolerance to the fetus (Burwick et al, 2021). This inflammation is tightly regulated to avoid excessive immune responses that may be harmful to embryo development. However, pregnancy complications associated with aberrant inflammation can create innate immune memory of an abnormal immunological pregnancy signature (Lodge-Tulloch et al, 2022). This immune maladaptation can be cross-generationally transferred via innate immune memory, increasing the risk for non-communicable diseases, including pregnancy complications, and cardiovascular and metabolic disorders in the offspring (Lodge-Tulloch et al, 2022).

## Perspectives and conclusions

The previously accepted paradigm that only parental genetic information is transmitted to offspring phenotypes has been recently challenged. Increasing evidence shows that beyond biological inheritance of the parental genome, an upper layer of regulation within epigenome is shaped after environmental exposures and can contribute to the phenotype of the next generations via epigenetic inheritance. The existence of mechanisms of epigenetic inheritance is now demonstrated in plants, invertebrates, mammals, and even humans. Environmental factors such as dietary changes, stress, and infections can induce epigenetic reprogramming of cells, modulating their expression patterns and phenotype. These epigenetic marks can be inherited across generations and modify offspring phenotype in different ways, either by contributing to its adaptation to environmental signals and increasing the survival or by promoting the development of diseases.

However, much remains to be learned about the mechanisms mediating transgenerational effects, and more studies are needed to increase the burden of evidence in the context of biological

---

**Box 1   Pending issues**

(i) Conduct in-depth causation studies in mammals, including micro-injection of oocytes or genetic manipulation of animal models to provide more robust evidence about the occurrence of transgenerational inheritance of epigenetic signatures induced by infection or inflammation.

(ii) Conduct more mechanistic studies in mammals and invertebrates of how different types of infection (bacterial, viral, fungal, or parasitic) or inflammation alter the epigenetic landscape, explore into the specific epigenetic processes and how these changes are transmitted across generations.

(iii) Investigate and discriminate the differences in the epigenetic mechanisms that mediate paternal versus maternal transgenerational epigenetic inheritance, providing more evidence for paternal contributions.

(iv) Evaluate the strength and persistence of the transgenerational effects by exploring beyond F3 generation.

(v) Investigate the mechanisms mediating the sex-specific differences of the transgenerational effects.

(vi) In humans, large-scale epidemiological studies with long-term follow-ups across multiple generations are necessary to assess the implications of transgenerational effects by infections or vaccination of the parents on health parameters and disease susceptibility, validate the findings, and rule out confounding factors.

plausibility and evolutionary adaptation, as well as health and disease implications (see also Box 1). Critical research gaps and confounding factors in terms of mechanisms involved, the various infections that induce transgenerational effects, and the strength of these effects needs to be investigated in future studies. Further research is needed for a better understanding of the transgenerational effects and their implications (Cime-Castillo et al, 2023; Fatemi et al, 1998; Gnambani et al, 2023; Gomes et al, 2023; He et al, 2022; He et al, 2023; Hernández López et al, 2014; López-Cervantes et al, 2022; Mednick et al, 1988; Mondotte et al, 2020; Norouzitallab et al, 2016; Nyangahu et al, 2020; Ory et al, 2022; Richter and Robling, 2013; Rodriguez-Andres et al, 2024; Roy et al, 2019; Sadd et al, 2005; Sengupta et al, 2024; Swaggerty et al, 2023; Xiu et al, 2019; Yue et al, 2013).

## Peer review information

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

## Acknowledgements

MGN was supported by an ERC Advanced Grant (#833247) and a Spinoza Grant of the Netherlands Organization for Scientific Research. Figures were created with BioRender.com.

## Author contributions

**Victoria M Spanou**: Software; Visualization; Writing—original draft; Writing—review and editing. **Theano P Andriopoulou**: Writing—original draft; Writing—review and editing. **Evangelos J Giamarellos-Bourboulis**: Supervision; Writing—review and editing. **Mihai G Netea**: Conceptualization; Supervision; Funding acquisition; Project administration; Writing—review and editing.

## Disclosure and competing interests statement

