## [Peer Review File · EMBO Molecular Medicine]

Improving the odds of survival: transgenerational effects of infections

Victoria Spanou, Theano Andriopoulou, Evangelos J. Giamarellos-Bourboulis, and Mihai Netea

Corresponding author(s): Victoria Spanou (vimaspanou@med.uoa.gr), Mihai Netea (Mihai.Netea@radboudumc.nl)

Review Timeline:

Submission Date:	10th Mar 24
Editorial Decision:	2nd Apr 24
Revision Received:	19th Dec 24
Accepted:	24th Dec 24

Editor: Zeljko Durdevic

Transaction Report:

2nd Apr 2024

Dear Dr. Spanou,

Thank you for the submission of your manuscript to EMBO Molecular Medicine. We have now received feedback from the two reviewers who agreed to evaluate your manuscript. As you will see from the reports below, the referees are positive about its interest and timeliness, however, they also raise important criticisms. Therefore, addressing the reviewers' concerns in full will be necessary for further considering the manuscript in our journal.

I would also like to ask you to amend the following:

- 1) Add up to 5 keywords.
- 2) Glossary: The glossary is meant to explain some of the terms used for laymen. Could you please identify terms that may need an "explanation"?
- 3) Pending issues: At the end of each article is a box highlighting issues that still need further studies and where research efforts should converge. Could you identify some pending issues?
 - 1) Figures: Please revise the Figure 3. The figure is overcrowded and the font and some of the schematic drawings are too small. We might send the article to our graphic artist who will re-draw the figures for style and clarity. Therefore, please ensure the information shown is scientifically accurate and upload the file as a PDF (or SVG, or EPS), PowerPoint or Keynote in which the labels and objects are still editable. For figures created using Adobe Illustrator, please send the Illustrator (.ai) file.
- 4) Rename "Competing interests" to "Disclosure Statement & Competing Interests": We updated our journal's competing interests policy in January 2022 and request authors to consider both actual and perceived competing interests. Please review the policy <https://www.embopress.org/competing-interests> and update your competing interests if necessary.
- 5) Please correct the reference citation in the text and reference list. In the text, a reference should be cited by author and year of publication. Include a space between a word and the opening parenthesis of the reference that follows. In the reference list, citations should be listed in alphabetical order. Where there are more than 10 authors on a paper, 10 will be listed, followed by "et al.". Please check "Author Guidelines" for more information.
<https://www.embopress.org/page/journal/17574684/authorguide#referencesformat>

You can submit your revised files by logging onto our online manuscript tracking system or simply follow this link:

Link Unavailable

I hope that the referees' comments do not prove too problematic to address and I look forward to reading your next version.

Yours sincerely,

Zeljko Durdevic

*** IMPORTANT INFORMATION ***

- 1) a .doc formatted version of the manuscript text (including Figure legends and tables)
- 2) Separate figure files
- 3) a letter INCLUDING the reviewer's reports and your detailed responses to their comments.

Also, and to save some time should your paper be accepted, please read below for additional information regarding some features of our research articles:

1) Glossary: EMBO Molecular Medicine articles will be accompanied by a glossary explaining some of the terms used for laymen. I identified the following:

_____, _____, _____

Could you please help us in identifying terms that may need an "explanation" other terms that we can add to the glossary.

2) For more information: This is a short list of related web links for further consultation by the readers. Could you identify some relevant ones? Examples are patient associations, OMIM related links, databases, authors websites, etc.

3) Pending issues: At the end of each article we will have a box highlighting issues that still need further studies and where research efforts should converge (we call this the Pending issues box). From my reading I would say:

but I can see there may be many more. Could you work on this as well?

4) Disclosure and competing interest statement: Please include a statement declaring any competing commercial interests in relation to your submitted work.

5) Please note that we now mandate that all corresponding authors list an ORCID digital identifier. This takes <90 seconds to complete. We encourage all authors to supply an ORCID identifier, which will be linked to their name for unambiguous name identification.

Currently, our records indicate that there is no ORCID associated with your account.

Please click the link below to provide an ORCID:

Link Not Available

-

Thank you,

Zeljko Durdevic

***** Reviewer's comments *****

Referee #1 (Remarks for Author):

This review manuscript, on transgenerational effects of infections, is timely and generally well written. There is a growing literature, and an urgent need for more research in this new field. However, there are various aspects of the manuscript that need to be improved and extended, as detailed below.

1. The authors should distinguish studies that only show correlation/association, from those that show causation/mechanism. This is particularly relevant to the potential mechanisms mediating paternal epigenetic inheritance. The one study [Reference 71] claiming to show a causative role for DNA methylation is not convincing, and requires independent replication. Furthermore, the epigenetic editing in the study in Reference 71 was not performed on male germ cells in vivo. In contrast, the evidence that non-coding RNAs contributed to paternal epigenetic inheritance is substantial. As well as the cited studies [e.g. Reference 53], the authors should cite other studies that show that sperm small and long non-coding RNAs can contribute to paternal epigenetic inheritance (via microinjection of fertilized oocytes), including:

Gapp K, et al. Implication of sperm RNAs in transgenerational inheritance of the effects of early trauma in mice. *Nat Neurosci*. 17(5):667-9. doi: 10.1038/nn.3695. Epub 2014 Apr 13.

Rodgers AB, et al. Transgenerational epigenetic programming via sperm microRNA recapitulates effects of paternal stress. *Proc Natl Acad Sci*. 112(44):13699-704. doi: 10.1073/pnas.1508347112. Epub 2015 Oct 19.

Zhang Y, et al. Dnmt2 mediates intergenerational transmission of paternally acquired metabolic disorders through sperm small non-coding RNAs. *Nat Cell Biol*. 20(5):535-540. doi: 10.1038/s41556-018-0087-2. Epub 2018 Apr 25.

Gapp K, et al. Alterations in sperm long RNA contribute to the epigenetic inheritance of the effects of postnatal trauma. *Mol Psychiatry*. 2020 Sep;25(9):2162-2174. doi: 10.1038/s41380-018-0271-6. Epub 2018 Oct 30.

Hoffmann LB, et al. Chronically high stress hormone levels dysregulate sperm long noncoding RNAs and their embryonic microinjection alters development and affective behaviours. *Mol Psychiatry*. 2023 Dec 19. doi: 10.1038/s41380-023-02350-2.

2. Related to the above points, the authors state in lines 124-125 - 'In particular, sperm DNA methylome and noncoding RNAs mediate these effects through the paternal germline'. The evidence that DNA methylome is causatively involved is less substantiated. It may be correct, but the key in vivo epigenetic editing experiments to test causation are yet to be performed.

3. It would be informative if the authors could include citations/references to the information summarised in their figures, particularly Figures 1 and 3. Whilst most of these are presumably cited in the main text, they also need to be cited in the relevant parts of the figures, so that the reader can easily find the primary evidence.

4. Figure 1. is titled 'Animal and plant groups in which transgenerational effects have been demonstrated'. The authors should be careful with their terminology, both in the figures and throughout the main text. Are these all 'transgenerational epigenetic inheritance' or just 'intergenerational epigenetic inheritance'?

5. Similarly, the authors should make a clearer distinction between maternal epigenetic inheritance and paternal epigenetic inheritance, which clearly involve distinct mechanisms.

6. The authors should generate one or more summary tables with previous publications and their key findings. There is a large literature and it needs to be integrated in a way to provide clarity for the reader. This could be according to host classes (and kingdoms down to species) of organisms, as well as maternal versus paternal, intergenerational versus transgenerational, and types of infection.

7. The authors do not adequately address the mechanistic issues. For example, how might different types of infections and immune activations cause epigenetic changes in germ cells? How might this alter offspring development, structure and function? This deserves more discussion, and at least one more figure to illustrate potential mechanisms, at molecular, cellular and systems levels.

8. The authors need to emphasise the importance of ruling out 'vertical transmission' in studies on epigenetic inheritance following infection. If a parent is simply passing the infection on to offspring, that is a completely different phenomenon, and biology.

9. The authors need to ensure that the key relevant literature is thoroughly discussed. For example, these highly relevant recent studies from the past year are worthy of discussion:

Gyawali S, et al. Maternal and paternal tuberculosis is associated with increased asthma and respiratory symptoms in their offspring: a study from Northern Europe. *Front Allergy*. 2023 Jun 8;4:1193141. doi: 10.3389/falgy.2023.1193141.

Cime-Castillo J, et al. The costs of transgenerational immune priming for homologous and heterologous infections with different serotypes of dengue virus in *Aedes aegypti* mosquitoes. *Front Immunol*. 2023 Dec 18;14:1286831. doi: 10.3389/fimmu.2023.1286831.

Rodriguez-Andres J, et al. Mosquito transgenerational antiviral immunity is mediated by vertical transfer of virus DNA sequences and RNAi. *iScience*. 2023 Nov 30;27(1):108598. doi: 10.1016/j.isci.2023.108598.

He Y, et al. Transgenerational epigenetic inheritance and immunity in chickens that vary in Marek's disease resistance. *Poult Sci*. 102(12):103036. doi: 10.1016/j.psj.2023.103036. Epub 2023 Aug 26.

Wibisono P, Sun J. Pathogen infection induces specific transgenerational modifications to gene expression and fitness in *Caenorhabditis elegans*. *Front Physiol*. 2023 Sep 22;14:1225858. doi: 10.3389/fphys.2023.1225858.

Gnambani EJ, et al. Infection of the malaria vector *Anopheles coluzzii* with the entomopathogenic bacteria *Chromobacterium anophelis* sp. nov. IRSSOUMB001 reduces larval survival and adult reproductive potential. *Malar J*. 2023 Apr 13;22(1):122. doi: 10.1186/s12936-023-04551-0.

Kim YH, et al. Ingestion of heat-killed pathogens confers transgenerational immunity to the pathogens via the vitellogenin-hypopharyngeal gland axis in honeybees. *Dev Comp Immunol*. 144:104709. doi: 10.1016/j.dci.2023.104709. Epub 2023 Apr 7.

Kleeman EA, et al. Paternal immune activation by Poly I:C modulates sperm noncoding RNA profiles and causes transgenerational changes in offspring behavior. *Brain Behav Immun*. 115:258-279. doi:10.1016/j.bbi.2023.10.005. Epub 2023 Oct 10.

Swaggerty CL, et al. Addition of a protected complex of biofactors and antioxidants to breeder hen diets confers transgenerational protection against *Salmonella enterica* serovar Enteritidis in progeny chicks. *Poult Sci*. 102(4):102531. doi: 10.1016/j.psj.2023.102531. Epub 2023 Jan 24.

10. Furthermore, is there no evidence from non-mammalian vertebrate species? This seems surprising, including from well-studied (including laboratory) species of birds, fish, amphibians, reptiles, etc. For example, in the studies/references listed in comment #6 above, there are a couple of studies on chickens/poultry.

11. There are various typographical (or other) errors that need correction, including:
Line 64 - 'advances in epigenetics support subtle quasi-Lamarckian model';

- Insert 'a' before 'subtle';
- Line 151 - correct 'repeteadly';
- Line 208 - 'trans-generational' (versus 'transgenerational' elsewhere in manuscript);
- Be consistent with use of hyphens in such words;
- Line 234 - 'For instance, male offspring from immune challenged; with LPS bumblebees of the species *Bombus terrestris* showed'
- Re-phrase;
- Line 624 - 'While only 2% of our DNA encodes for proteins, the rest of 98%';
- Change to 'the rest (98%)'
- Line 75 - 'mammals, and even humans';
- Change to 'vertebrates including mammals, and even humans'.

Referee #2 (Remarks for Author):

- The overall topic of the review is timely and important and should be of interest to an audience larger than those solely interested in epigenetic inheritance. Several aspects would greatly improve the review and help magnify its impact. In order:
- Rewrite the abstract to be more descriptive of the actual content of the review
 - In the first section, include a working definition of epigenetics and inter, multi, and transgenerational inheritance.
 - Move the section on epigenetics placed at the end to the beginning as this is needed to understand the rest of the review.
 - Clarify how selection can act on epialleles (line 80). Give an example.
 - Line 94-97, there are many examples of deleterious effects, from "lower" organisms to humans. The issue is more of the opposite, not many studies have shown a beneficial role for TEI.
 - Line 104, make it clear that reprogramming is of DNA methylation and is mostly a mammalian phenomenon.
 - Paragraph starting on line 146, add references
 - A VERY LARGE OMISSION in section starting on line 144: there was no mention of Coleen Murphy's work.
 - In the same section, there is an overall lack of distinction in the text between what is a DIRECT effect on the offspring from the infection, what is truly passed intergenerationally, and what is multi and trans. This should be very explicit throughout the review. A direct impact of infection of the offspring would not be inheritance.
 - Section that begins on line 493, as mentioned earlier, this section belongs at/near the front of the manuscript.

We would like to thank both reviewers for their thorough examination of our manuscript and we appreciate their constructive suggestions and comments that helped us improve the content of the manuscript.

Referee #1 (Remarks for Author): This review manuscript, on transgenerational effects of infections, is timely and generally well written. There is a growing literature, and an urgent need for more research in this new field. However, there are various aspects of the manuscript that need to be improved and extended, as detailed below.

Thank you very much for the positive comment! I hope that the revised manuscript is improved and expanded to include all the new and significant research in this field.

1. The authors should distinguish studies that only show correlation/association, from those that show causation/mechanism. This is particularly relevant to the potential mechanisms mediating paternal epigenetic inheritance. The one study [Reference 71] claiming to show a causative role for DNA methylation is not convincing and requires independent replication. Furthermore, the epigenetic editing in the study in Reference 71 was not performed on male germ cells in vivo. As well as the cited studies [e.g. Reference 53], the authors should cite other studies that show that sperm small and long non-coding RNAs can contribute to paternal epigenetic inheritance (via microinjection of fertilized oocytes), including:

1. Gapp K, et al. Implication of sperm RNAs in transgenerational inheritance of the effects of early trauma in mice. *Nat Neurosci.* 17(5):667-9. doi: 10.1038/nn.3695. Epub 2014 Apr 13.
2. Rodgers AB, et al. Transgenerational epigenetic programming via sperm microRNA recapitulates effects of paternal stress. *Proc Natl Acad Sci.* 112(44):13699-704. doi: 10.1073/pnas.1508347112. Epub 2015 Oct 19.
3. Zhang Y, et al. Dnmt2 mediates intergenerational transmission of paternally acquired metabolic disorders through sperm small non-coding RNAs. *Nat Cell Biol.* 20(5):535-540. doi: 10.1038/s41556-018-0087-2. Epub 2018 Apr 25.
4. Gapp K, et al. Alterations in sperm long RNA contribute to the epigenetic inheritance of the effects of postnatal trauma. *Mol Psychiatry.* 2020 Sep;25(9):2162-2174. doi: 10.1038/s41380-018-0271-6. Epub 2018 Oct 30.
5. Hoffmann LB, et al. Chronically high stress hormone levels dysregulate sperm long noncoding RNAs and their embryonic microinjection alters development and affective behaviours. *Mol Psychiatry.* 2023 Dec 19. doi: 10.1038/s41380-023-02350-2.

Answer. We thank the reviewer for the useful suggestion. Indeed, the separation of the causation and the correlation/association studies is very important to provide more clarity. As suggested, the proposed publications have been added to the revised manuscript. The discrimination of the causation studies has been highlighted in a new paragraph in which we discuss the studies associated with causally paternal epigenetic inheritance.

2. Related to the above points, the authors state in lines 124-125 - 'In particular, sperm DNA methylome and noncoding RNAs mediate these effects through the paternal germline'. The evidence that DNA methylome is causatively involved is less substantiated. It may be correct, but the key in vivo epigenetic editing experiments to test causation are yet to be performed.

Answer. This point has now been corrected according to the reviewer's suggestion.

3. It would be informative if the authors could include **citations/references** to the information summarized in their figures, particularly **Figures 1 and 3**. Whilst most of these are presumably cited in the main text, they also need to be cited in the relevant parts of the figures, so that the reader can easily find the primary evidence.

Answer. As suggested by the reviewer, we added the relevant references to the figures. To improve even further the accessibility of information, we propose to provide a Supplementary Table that summarizes the studies on trans-generational effects of infections and their main findings.

4. **Figure 1**. is titled 'Animal and plant groups in which transgenerational effects have been demonstrated'. The authors should be **careful with their terminology**, both in the figures and throughout the main text. Are these all '**transgenerational epigenetic inheritance**' or just '**intergenerational epigenetic inheritance**'?

Answer. The figure 1 includes animals and plants in which intergenerational or transgenerational epigenetic inheritance, or both, have been demonstrated. The legend title has now been modified to be more general: inter- and transgenerational. In the main text, these two terms have been revised and corrected when needed.

5. Similarly, the authors should make a **clearer distinction between maternal epigenetic inheritance and paternal epigenetic inheritance**, which clearly involve distinct mechanisms.

Answer. We agree with the reviewer. In the revised manuscript, the inheritance through the different germlines was organized in separate paragraphs with an effort to achieve the best clarity.

6. The authors should **generate one or more summary tables** with previous publications and their key findings. There is a large literature, and it needs to be integrated in a way to provide clarity for the reader. **This could be according to host classes (and kingdoms down to species) of organisms, as well as maternal versus paternal, intergenerational versus transgenerational, and types of infection.**

Answer. As suggested by the reviewer, we constructed a summary table that contains the information requested (host classes, maternal vs paternal, intra-, inter- or transgenerational, types of infection/immune activation, key findings) with all the publications. The table is very large, and we propose to be added as Supplementary

Table to the publication. However, if the editors consider that the Table should be part of the main body of the manuscript, we agree of course.

7. The authors do not adequately address the **mechanistic issues**. For example, how might different types of infections and immune activations cause epigenetic changes in germ cells? How might this alter offspring development, structure and function? This deserves **more discussion**, and **at least one more figure** to illustrate potential mechanisms, at molecular, cellular and systems levels.

Answer. As suggested by the reviewer, a new figure has been generated describing the mechanisms suggested to be involved, by the limited causation studies in mammals. Due to the limited number of studies in the field, the mechanistic link between the induction of the epigenetic changes, the transfer through the germline and the modification of the progeny phenotype is still unclear. Thus, this gap of knowledge has been now highlighted indicating the need of further studies exploring the epigenetic impact of the different types of infections in the germline cells and their effects in the offspring of multiple generations.

8. The authors need to **emphasize the importance of ruling out 'vertical transmission'** in studies on epigenetic inheritance following **infection**. If a parent is simply passing the infection on to offspring, that is a completely different phenomenon, and biology.

Answer. This is a very important point. This clarification has now been added to the manuscript to emphasize the importance of confounding factors in the studies of epigenetic inheritance, in the section 6.1.

9. The authors need to ensure that the key relevant literature is thoroughly discussed. For example, these highly relevant recent studies from the past year are worthy of discussion:

1. Gyawali S, et al. **Maternal and paternal tuberculosis is associated with increased asthma and respiratory symptoms in their offspring**: a study from Northern Europe. *Front Allergy*. 2023 Jun 8;4:1193141. doi: 10.3389/falgy.2023.1193141.
2. Cime-Castillo J, et al. The costs of transgenerational immune priming for homologous and heterologous infections with **different serotypes of dengue virus in Aedes aegypti mosquitoes**. *Front Immunol*. 2023 Dec 18;14:1286831. doi: 10.3389/fimmu.2023.1286831.
3. Rodriguez-Andres J, et al. **Mosquito transgenerational antiviral immunity is mediated by vertical transfer of virus DNA sequences and RNAi**. *iScience*. 2023 Nov 30;27(1):108598. doi: 10.1016/j.isci.2023.108598.
4. He Y, et al. **Transgenerational epigenetic inheritance and immunity in chickens that vary in Marek's disease resistance**. *Poult Sci*. 102(12):103036. doi: 10.1016/j.psj.2023.103036. Epub 2023 Aug 26.
5. Wibisono P, Sun J. Pathogen infection induces specific transgenerational modifications to gene expression and fitness in **Caenorhabditis elegans**. *Front Physiol*. 2023 Sep 22;14:1225858. doi: 10.3389/fphys.2023.1225858.

6. Gnambani EJ, et al. Infection of the malaria vector **Anopheles coluzzii** with the entomopathogenic bacteria **Chromobacterium** anophelis sp. nov. IRSSSOUMB001 reduces larval survival and adult reproductive potential. *Malar J.* 2023 Apr 13;22(1):122. doi: 10.1186/s12936-023-04551-0.
7. Kim YH, et al. Ingestion of heat-killed pathogens confers transgenerational immunity to the pathogens via the vitellogenin-hypopharyngeal gland axis in **honeybees**. *Dev Comp Immunol.* 144:104709. doi: 10.1016/j.dci.2023.104709. Epub 2023 Apr 7.
8. Kleeman EA, et al. Paternal immune activation by **Poly I:C** modulates sperm noncoding RNA profiles and causes transgenerational changes in **offspring behavior**. *Brain Behav Immun.* 115:258-279. doi:10.1016/j.bbi.2023.10.005. Epub 2023 Oct 10.
9. Swaggerty CL, et al. Addition of a protected complex of biofactors and antioxidants to **breeder hen** diets confers transgenerational protection against **Salmonella** enterica serovar Enteritidis in progeny chicks. *Poult Sci.* 102(4):102531. doi: 10.1016/j.psj.2023.102531. Epub 2023 Jan 24.

Answer. All the suggested literature has now been added to the manuscript. Thank you very much for the proposal of these relevant and important studies.

10. Furthermore, is there no evidence from **non-mammalian vertebrate species**? This seems surprising, including from well-studied (including laboratory) species of **birds, fish, amphibians, reptiles**, etc. For example, in the studies/references listed in comment #6 above, there are a couple of studies on chickens/poultry.

Answer. While the evidence of transgenerational epigenetic inheritance in vertebrates is relatively scarce, there are indeed a number of studies in non-mammalian species that argue for trans-generational effects mainly in fish. However, most studies, especially in zebrafish, focus on exposure to xenobiotics while studies related to the transmission of immune or pathogenic-induced epigenetic marks are lacking. Few data about infection-induced epigenetic changes in fish at an intragenerational level are documented. Also, in the present review we wanted to focus on the mammalian transgenerational effects, as otherwise the text would become much broader, and we could not cover the subject exhaustively due to text limitation. Importantly, the different mechanisms that mediate the epigenetic inheritance in these species could not be discussed adequately. We therefore added information on non-mammalian vertebrates in the additional Table.

11. There are various **typographical** (or other) errors that need correction, including:
 Line 64 - 'advances in epigenetics support subtle quasi-Lamarckian model'; - Insert 'a' before 'subtle';
 Line 151 - correct 'repeatedly';
 Line 208 - 'trans-generational' (versus 'transgenerational' elsewhere in manuscript); - Be consistent with use of hyphens in such words;
 Line 234 - 'For instance, male offspring from immune challenged; with LPS bumblebees of the species *Bombus terrestris* showed' - Re-phrase;
 Line 624 - 'While only 2% of our DNA encodes for proteins, the rest of 98%'; - Change to 'the rest (98%)'

Line 75 - 'mammals, and even humans'; - Change to 'vertebrates including mammals, and even humans'.

Answer. We would like to thank the reviewer for her/his thorough and detailed reading of our manuscript. The typos and other aspects raised by the reviewer were corrected in the revised version of the manuscript.

Referee #2 (Remarks for Author): The overall topic of the review is timely and important and should be of interest to an audience larger than those solely interested in epigenetic inheritance. Several aspects would greatly improve the review and help magnify its impact. In order:

Answer. Thank you for your kind comments! We believe the revised manuscript has been significantly improved according to your suggestions.

1. **Rewrite the abstract** to be more descriptive of the actual content of the review

Answer. The abstract has been modified to be more specific and it described now better the content of the review, as per your suggestion.

2. In the first section, include a working definition of epigenetics and **inter, multi,** and **transgenerational** inheritance.

Answer. Definitions of the terms intra-, multi, inter-, and transgenerational inheritance have now been added at the beginning of the manuscript to provide clarity and facilitate reading.

3. **Move the section** on epigenetics placed at the end to the **beginning** as this is needed to understand the rest of the review.

Answer. The section of epigenetics is transferred to the beginning of the review as per your suggestion.

4. **Clarify how selection can act on epialleles** (line 80). Give an example.

Answer. On line 80 of the revised manuscript, an example of how selection acts on plants' epialleles is given to improve the understanding.

5. Line 94-97, there are many examples of deleterious effects, from "lower" organisms to humans. **The issue is more of the opposite, not many studies have shown a beneficial role for TEI.**

Answer. We would like to thank the reviewer for this critical comment. Indeed, there are a lot of studies demonstrating deleterious effects and trade-offs or costs of TEI. This sentence has been corrected accordingly.

6. Line 104, **make it clear that reprogramming is of DNA methylation and is mostly a mammalian phenomenon.**

Answer. Thank you very much for pointing this out, an explanation has been added to provide more clarity. This point has been added to the manuscript.

7. Paragraph starting on line 146, **add references**

Answer. The relevant references have been added as per your indication.

8. A VERY LARGE OMISSION in section starting on line 144: there was no mention of Coleen Murphy's work.

Answer. The relevant work of Professor Coleen T. Murphy has now been added to the manuscript in a separate paragraph regarding the research in C. elegans. Thank you for the suggestion.

9. In the same section, there is an **overall lack of distinction** in the text between **what is a DIRECT effect** on the offspring from the infection, **what is truly passed intergenerationally, and what is multi and trans.** This should be very explicit throughout the review. **A direct impact of infection of the offspring would not be inheritance.**

Answer. We emphasized the critical need to rule out the “vertical transmission” following infection in studies of epigenetic inheritance through maternal germline in the section 6.1.

A new figure with the causative studies and an extended summary table with discrimination of the intergenerational and transgenerational effects through the female and male germline have been generated to provide more clarity to the manuscript.

10. Section that begins on line 493, as mentioned earlier, this section belongs at/near the front of the manuscript.

Answer. We would like to thank the reviewer for the suggestion, this section is now transferred to the beginning of the manuscript.

Given that the main focus is on the mammal species, the following publication has also been added.

López-Cervantes JP, Shigdel R, Accordini S, Mustafa T, Bertelsen RJ, Makvandi-Nejad S, Lerm M, Horsnell W, Svanes C (2022) Parental TB associated with offspring asthma and rhinitis. Int J Tuberc Lung Dis 26: 544-549

24th Dec 2024

Dear Dr. Spanou,

We are pleased to inform you that your manuscript is accepted for publication and is now being sent to our publisher to be included in the next available issue of EMBO Molecular Medicine.

Your manuscript will be processed for publication by EMBO Press. It will be copy edited and you will receive page proofs prior to publication.

There is no charge for this Review Article. However, in a few weeks, when you are contacted to sign your license agreement and review the article proofs, you will need to enter a token into the appropriate field in the Springer Nature Author Services system. Please note that we will provide the token in a separate letter. Be aware that, due to the holiday season, we anticipate a delay in processing your manuscript.
